# Detection of acute dengue virus infection, with and without concurrent malaria infection, in a cohort of febrile children in Kenya, 2014–2019, by clinicians or machine learning algorithms

David M. Vu[1]*, Amy R. Krystosik[1], Bryson A. Ndenga[2], Francis M. Mutuku[3], Kelsey Ripp[4], Elizabeth Liu[1], Carren M. Bosire[5], Claire Heath[1], Philip Chebii[6], Priscilla Watiri Maina[6], Zainab Jembe[7], Said Lipi Malumbo[6], Jael Sagina Amugongo[6], Charles Ronga[2], Victoria Okuta[8], Noah Mutai[2], Nzaro G. Makenzi[5], Kennedy A. Litunda[5], Dunstan Mukoko[9], Charles H. King[10], A. Desiree LaBeaud[1]

**1** Department of Pediatrics, Division of Infectious Diseases, Stanford University School of Medicine, Stanford, California, United States of America, **2** Centre for Global Health Research, Kenya Medical Research Institute, Kisumu, Kenya, **3** Department of Environment and Health Sciences, Technical University of Mombasa, Mombasa, Kenya, **4** University of Global Health Equity, Butaro, Rwanda, **5** Department of Pure and Applied Sciences, Technical University of Mombasa, Mombasa, Kenya, **6** Vector-Borne Diseases Unit, Msambweni County Referral Hospital, Msambweni, Kwale, Kenya, **7** Vector-Borne Diseases Unit, Diani Health Center, Ukunda, Kwale, Kenya, **8** Paediatric Department, Obama Children's Hospital, Jaramogi Oginga Odinga Referral Hospital, Kisumu, Kenya, **9** Vector-Borne Diseases Unit, Ministry of Health, Nairobi, Kenya, **10** Department of Pathology, Center for Global Health and Diseases, Case Western Reserve University School of Medicine, Cleveland, Ohio, United States of America

* davidvu@stanford.edu

## Abstract

Poor access to diagnostic testing in resource limited settings restricts surveillance for emerging infections, such as dengue virus (DENV), to clinician suspicion, based on history and exam observations alone. We investigated the ability of machine learning to detect DENV based solely on data available at the clinic visit. We extracted symptom and physical exam data from 6,208 pediatric febrile illness visits to Kenyan public health clinics from 2014–2019 and created a dataset with 113 clinical features. Malaria testing was available at the clinic site. DENV testing was performed afterwards. We randomly sampled 70% of the dataset to develop DENV and malaria prediction models using boosted logistic regression, decision trees and random forests, support vector machines, naïve Bayes, and neural networks with 10-fold cross validation, tuned to maximize accuracy. 30% of the dataset was reserved to validate the models. 485 subjects (7.8%) had DENV, and 3,145 subjects (50.7%) had malaria. 220 (3.5%) subjects had co-infection with both DENV and malaria. In the validation dataset, clinician accuracy for diagnosis of malaria was high (82% accuracy, 85% sensitivity, 80% specificity). Accuracy of the models for predicting malaria diagnosis ranged from 53–69% (35–94% sensitivity, 11–80% specificity). In contrast, clinicians detected only 21 of 145 cases of DENV (80% accuracy, 14% sensitivity, 85% specificity). Of the six models, only logistic regression identified any DENV case (8 cases, 91% accuracy,

**Data Availability Statement:** The dataset "Detection of acute dengue virus infection, with and

without concurrent malaria infection, at Kenyan clinics in a cohort of acutely febrile children, 2014-2019, by clinicians or machine learning" is available for download on datadryad.org (https://datadryad.org/stash/share/lsdTkRyOoVajK6WLYpAaBNmMJwnNM9_p87wfWeKXI2o).

**Funding:** This work was supported by the National Institute of Allergy and Infectious Diseases at the National Institutes of Health [NIH K23 AI127909] to D.M.V. and [NIH R01 AI102918] to A.D.L.; the Stanford Maternal & Child Health Research Institute, Lucille Packard Foundation for Children's Health, and Spectrum (the Stanford Center for Clinical & Translational Research & Education) [KL2 TR 001083, UL1 TR 001085] to D.M.V. The funders of the study had no role in study design, data collection, data analysis, data interpretation, or writing of the report, or in the decision to submit the paper for publication.

**Competing interests:** The authors have declared that no competing interests exist.

5.5% sensitivity, 98% specificity). Without diagnostic testing, interpretation of clinical findings by humans or machines cannot detect DENV at 8% prevalence. Access to point-of-care diagnostic tests must be prioritized to address global inequities in emerging infections surveillance.

## Introduction

Poor access to diagnostic testing is an important driver of global health inequity in emerging infections surveillance, and has been identified by clinicians practicing in resource restricted settings in Kenya as a barrier to the accurate diagnosis and treatment of infections [1]. In areas of high malaria transmission, malaria diagnosis can mask the consideration and recognition of alternate or concurrent causes of febrile disease, such as dengue virus (DENV) co-infection, simply based on the pursuit of diagnostic parsimony (i.e. "Occam's razor" or the notion that the simplest diagnosis is correct) [2]. Indeed, co-infection with DENV and malaria is increasingly being reported around the world [3,4] and specifically in Kenya [5]. Yet, lack of routine surveillance for DENV in sub-Saharan Africa leads to continued underestimation of its disease burden [6–10], driving a vicious cycle; Under-recognition of DENV [1] impairs detection of outbreaks, which misinforms decision-making regarding needed public health interventions and perpetuates poor public awareness of DENV. Underdiagnosis of DENV also facilitates misclassification of disease due to DENV/malaria co-infection as malaria infection alone [11].

Inequitable access to diagnostic testing for DENV has prompted investigators to develop clinical prediction models, such as decision trees, that utilize a combination of symptoms and physical exam findings plus laboratory characterization of hematologic parameters, including platelet count and hematocrit, to either diagnose DENV or classify DENV severity [12,13]. However, poor and inconsistent access to such laboratory tests in most Kenyan public health facilities render such aids unusable [14]. In the absence of available testing, healthcare providers must rely on the reported symptoms and physical exam observations to make management decisions. The focus of this report is to describe the clinical manifestations of DENV and malaria solo-infection, and DENV/malaria co-infection in children living in Kenya in regions of high year-round transmission of *P. falciparum* malaria, and to investigate whether differences in disease manifestations observed by clinicians are sufficient for the identification of DENV infection in absence of diagnostic testing. We hypothesized that machine learning algorithms could predict DENV infection in our cohort based on presence or absence of different clinical manifestations.

## Materials and methods

### Ethics statement

This prospective cohort study (NIH R01 AI102918, P.I. LaBeaud) was conducted between January 1, 2014 and June 30, 2019, with approval from the ethics committee of the Kenya Medical Research Institute (SSC 2611) and Stanford University School of Medicine (IRB-31488) [5–7]. Additional information regarding the ethical, cultural, and scientific considerations specific to inclusivity in global research is included in the Supporting Information (S3 Text). We recruited all persons aged 1–17 years with an acute febrile illness (AFI), defined as either reported history of fever or having a temperature of 38˚C at the time of examination, who presented to any of the four clinic sites during the study period: Chulaimbo (rural setting) and

Kisumu (urban setting) were located on the western border of Kenya, while Msambweni (rural setting) and Ukunda (urban setting) were located on the coast of Kenya. Written informed consent was obtained from the parent or guardian of each participant under 18 years of age.

Clinical officers evaluated and treated patients under the auspices of the Kenya Ministry of Health, independent of participation in the study. Data were recorded by aides onto tablet computers and secured on REDCap (Vanderbilt University, Nashville, TN). Symptoms as reported by the subject and/or guardian, and physical examination findings as observed by the clinician, were entered as multiple-select choices and as text entries to minimize survey bias. Blood was sampled by venipuncture. We recorded the clinician's diagnosis, treatment recommendations, and whether the child was admitted to the hospital. The subject was asked to return for a follow up (convalescent) study visit four weeks later, at which point these procedures were repeated.

We administered a demographic survey (S1 Text) during the initial visit. We created a composite wealth index to assess socioeconomic status, as previously described [15]. We measured height and weight, and calculated height-for-age z scores (HAZ) and body-mass index-for-age z scores (BAZ) based on World Health Organization (WHO) standards [16] as indicators of growth and nutritional status using the R package "zscorer" [17]. At both the initial and follow up visits, we administered the PedsQL™ 4.0 SF15 health-related quality of life (HrQoL) survey [18], which we translated to Kiswahili and Dhuluo (S2 Text), to explore its utility as an assessment tool to screen for functional disabilities associated with AFIs.

Blood samples were tested for malaria at the point-of-care by light microscopy or RDT (SD Bioline Malaria Antigen Pf, Standard Diagnostics, Korea) per routine clinical practice. Malaria infection was defined by the presence of parasites on peripheral blood smears or by having a positive malaria RDT if smear results were unavailable. For our study, a single operator re-examined available blood smears to identify species based on parasite morphology. Testing for DENV RNA by RT-PCR was performed after the visit [5,19]. Anti-DENV serum IgG was measured by ELISA [7,8] that used a mixture of inactivated DENV-1 virus strain Hawaii, DENV-2 strain New Guinea C, DENV-3 strain H87 and DENV-4 strain H421 (American Type Culture Collection). DENV infection was defined by positive RT-PCR or by anti-DENV IgG seroconversion from negative at the acute visit to positive at the 1-month convalescent visit.

The *a priori* analysis plan investigated associations between DENV or malaria infection and reported symptoms and physical examination observations as well as sex, age, HAZ, BAZ, study site, geographic location (west vs. coast), and population density (urban vs. rural). Only subjects who had test results for both DENV and malaria were included in the complete case analysis. Subjects lost to follow up were excluded from the complete case analysis. Agreement between clinician diagnosis and laboratory diagnosis was characterized using Cohen's Kappa. Recorded symptoms and physical exam findings were coded manually by a single operator into individual binary clinical features. Group-wise comparisons were conducted using R programming language (version 4.0.2) [20]. Non-parametric Kruskal-Wallis H test and Student T test were used for continuous data, as appropriate. Chi-square was used for categorical features. Between strata comparisons were not performed for features with frequencies less than five, considered sparse data. We anticipated comparing the ability of up to 25 clinical features to predict infection class and used Bonferroni correction to set $\alpha = 0.002$ to guide selection of features to be included in the planned statistical models. We applied this alpha to all comparisons to minimize the risk of false discovery. Odds ratios were calculated by exponentiating the coefficient for each strata using multinomial logistic regression (R package "nnet") [21].

To explore whether the clinical features could be used to classify DENV or malaria solo-infection, or DENV/malaria co-infection, we applied six supervised machine learning algorithms to predict infection stratum as the primary outcome of interest. Classification and

regression trees were used due to the implicit feature selection and interpretability. Random forest was used as an ensemble technique to address overfitting by classification and regression trees. Similarly, we chose logistic regression with boosting. Support vector machine was chosen for its robustness. Naïve Bayes was chosen as an example of a probabilistic classifier. And multi-layer perceptron was chosen to represent artificial neural networks as an example of deep learning [22]. The R language package "missForest" [23] was used to impute missing age data. The R package "caret" version 6.0-90fa [24] was used to randomly partition 70% of the subjects in the complete cases dataset to develop the models, while the remaining 30% was reserved to validate the models. For model development, we used 10-fold cross-validation and set the same starting seed. The de-identified dataset is available at https://doi.org/10.5061/dryad.rn8pk0pg1.

## Results

34 of the 7,543 enrolled subjects did not meet the definition for AFI and were excluded from the analysis (Fig 1). Among the remaining 7,509 subjects, 3,585 (47.7%) were female. There were no differences in sex distribution by site, region, or urban/rural classifications (Table 1). However, participants in Ukunda were older (median age 7.9 years compared to 4.8 years for the entire cohort, Table 1), contributing to differences in age observed between urban and rural sites, and between western and coastal sites (S1 Table). The median HAZ for Msambweni (-1.0, Table 1) was lower than the overall HAZ (-0.8) which contributed to the lower HAZ observed in the coastal compared to western sites (-0.8 vs -0.6, respectively, S1 Table), and rural compared to urban sites (-0.9 vs -0.7, respectively). The median BAZ also was lower in the coastal compared to western sites (-0.9 vs -0.3, respectively, S1 Table). Wealth index was higher in western compared to coastal sites (3 vs 2, respectively, S1 Table). Taken together, the differences highlight age, growth, nutrition, and wealth as possible confounding variables.

452 subjects were lost to follow up. The median interval between acute and convalescent visits was 31 (IQR 28, 35) days. 821 of the remaining 7,057 subjects were missing DENV test results due to insufficient sample volumes. 490 (7.9%) of the 6,236 subjects with DENV test results were positive. Yearly incidence varied, however, DENV cases were detected during 48 of the 66 months of sample collection, including months outside recognized outbreak periods (Fig 2).

3,757 (50.2%) of the 7,481 subjects with malaria test results were positive. *Plasmodium* species could be identified based on morphology by light microscopy for 2,642 subjects. 2,613

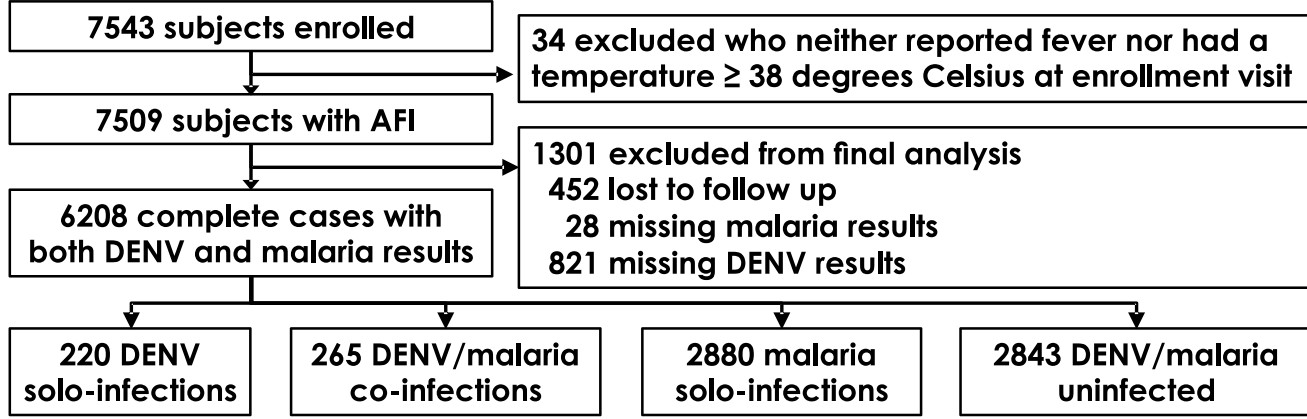

**Fig 1. Study enrollment and selection of complete cases.** Complete cases were defined as having both malaria and DENV test results.

**Table 1. Characteristics of all enrolled subjects with acute febrile illness[a] by study site.**

| | Total | By site | | | | $p^b$ (SMD[c]) |
| --- | --- | --- | --- | --- | --- | --- |
| | | Kisumu | Chulaimbo | Ukunda | Msambweni | |
| Enrolled, n | 7,509 | 1,711 | 833 | 2,557 | 2,408 | |
| Female, n (%) | 3,585 (47.7) | 811 (47.4) | 399 (47.9) | 1,228 (48.0) | 1,147 (47.6) | 0.98 (NA) |
| Age, median years (IQR) | 4.8 (2.9, 8.4) | 3.6 (2.0, 4.9) | 5.0 (3.0, 8.3) | 7.9 (5.1, 11.4) | 3.9 (2.3, 5.9) | <0.001 (0.69) |
| Height-for-age, median z (IQR) | -0.8 (-1.7, 0.2) | -0.7 (-1.7, 0.2) | -0.4 (-1.3, 0.5) | -0.7 (-1.6, 0.2) | -1.0 (-1.9, -0.1) | <0.001 (0.19) |
| BMI-for-age, median z (IQR) | -0.7 (-1.5, 0.2) | -0.2 (-1.1, 0.8) | -0.4 (-1.2, 0.4) | -1.0 (-1.9, -0.2) | -0.7 (-1.6, 0.2) | <0.001 (0.34) |
| Wealth index, median (IQR) | 2 (1, 3) | 3 (2, 3) | 2 (1, 3) | 2 (1, 3) | 2 (1, 3) | <0.001 (0.36) |

[a] Acute febrile illness defined as report of fever as symptom or recorded temperature ≥38 degrees Celsius at the initial clinic visit.

[b] Categorical variables tested by chi-square, and continuous variable tested using the Kruskal-Wallis H test.

[c] Standard mean differences (SMD) available for continuous variables only.

Abbreviations: SMD, standard mean difference; IQR, interquartile range; BMI, body mass index.

(98.9%) were identified as *P. falciparum*, consistent with previous reports that most of the malaria in Kenya is due to *P. falciparum* [25]. Other species that were identified included *P. malariae* (n = 30, 9 co-infected with *P. falciparum*) and *P. ovale* (n = 9, 1 co-infected with *P. falciparum*).

6,208 (83%) subjects had test results for both DENV and malaria and comprised the complete cases cohort (Fig 1). The enrolled and complete case cohorts were comparable with regards to age, sex, DENV incidence, malaria incidence, and whether the child was referred by the clinician to be hospitalized (*p*≥0.1, S2 Table). Differences in HAZ and BAZ did not reach our prespecified Bonferroni-adjusted α = 0.002. Of the 485 complete cases who had DENV, 265 (54.6%) also had malaria (DENV/malaria co-infection). DENV/malaria co-infections accounted for 8.4% of all malaria infections. No between-strata differences were detected for sex, growth (HAZ scores) or nutritional indicators (BAZ scores). However, differences in age and wealth index were observed (*p*<0.001, Table 2, see S3 Table for additional supporting statistics). The incidences of DENV and malaria were higher at the western sites (in particular,

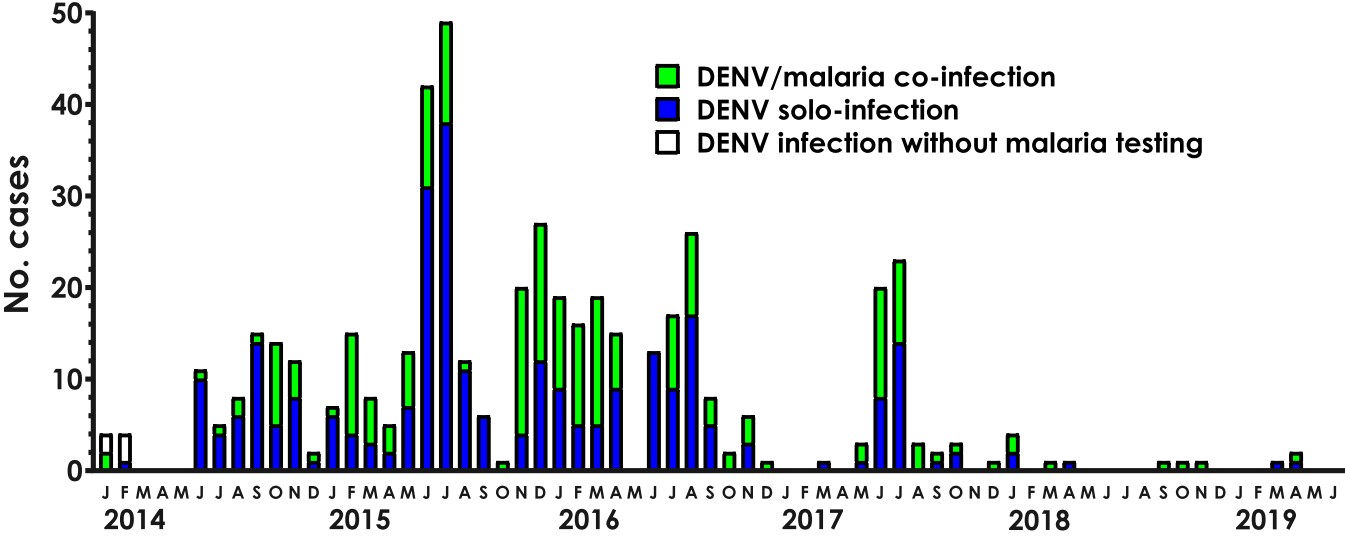

**Fig 2. DENV epidemic curve.** Month and year of 490 cases of DENV infection. DENV solo-infection, blue. DENV/malaria co-infection, green. DENV case missing malaria testing, unshaded.

**Table 2. Characteristics of complete cases by infection strata.**

| | Total | DENV/malaria co-infection | DENV solo-infection | malaria solo-infection | DENV/malaria uninfected | $p^a$ (SMD$^b$) |
|---|---|---|---|---|---|---|
| n (% of total cases) | 6208 (100) | 265 (4.3) | 220 (3.5) | 2880 (46.4) | 2843 (45.8) | NA |
| Female, n (% of stratum) | 2991 (48.2) | 123 (46.4) | 97 (44.1) | 1410 (49.0) | 1361 (47.9) | 0.46 (NA) |
| Age, median years (IQR) | 4.8 (2.9, 8.3) | 4.6 (3.0, 7.0) | 4.3 (2.3, 6.6) | 5.1 (3.3, 8.7) | 4.4 (2.5, 7.9) | <0.001 (0.015) |
| Height-for-age, median z-score (IQR) | -0.8 (-1.7, 0.1) | -0.8 (-1.8, 0.2) | -0.8 (-1.6, 0.3) | -0.8 (-1.7, 0.1) | -0.8 (-1.7, 0.2) | 0.36 (0.06) |
| BMI-for-age, median z-score (IQR) | -0.7 (-1.5, 0.2) | -0.5 (-1.2, 0.2) | -0.5 (-1.5, 0.2) | -0.7 (-1.5, 0.3) | -0.7 (-1.7, 0.2) | 0.003 (0.11) |
| Wealth index, median (IQR) | 2 (1, 3) | 2 (1, 3) | 3 (2, 3) | 2 (1, 3) | 2 (1, 3) | <0.001 (0.24) |
| Admitted to hospital, n (% of stratum) | 215 (3.5) | 20 (7.5) | 9 (4.1) | 127 (4.4) | 61 (2.1) | <0.001 (NA) |
| By study site | | | | | | |
| Kisumu, n (% of site total) | 1498 (100) | 77 (5.1) | 102 (6.8) | 570 (38.1) | 749 (50.0) | |
| Chulaimbo, n (% of site total) | 770 (100) | 120 (15.6) | 30 (3.9) | 518 (67.3) | 102 (13.2) | <0.001 (NA) |
| Ukunda, n (% of site total) | 1991 (100) | 25 (1.3) | 29 (1.5) | 985 (49.5) | 952 (47.8) | |
| Msambweni, n (% of site total) | 1949 (100) | 43 (2.2) | 59 (3.0) | 807 (41.4) | 1040 (53.3) | |
| By geography | | | | | | |
| West, n (% of region total) | 2268 (100) | 197 (8.7) | 132 (5.8) | 1088 (48.0) | 851 (37.5) | <0.001 (NA) |
| Coast, n (% of region total) | 3940 (100) | 68 (1.7) | 88 (2.2) | 1792 (45.5) | 1992 (50.6) | |
| By rural/urban | | | | | | |
| Urban, n (% of urban total) | 3489 (100) | 102 (2.9) | 131 (3.8) | 1555 (44.6) | 1701 (48.8) | <0.001 (NA) |
| Rural, n (% of rural total) | 2719 (100) | 163 (6.0) | 89 (3.3) | 1325 (48.7) | 1142 (42.0 | |

$^a$ Categorical variables tested by chi-square (see S3 Table for details), and continuous variables tested using the Kruskal-Wallis H test.

$^b$ Standard mean differences (SMD) available for continuous variables only.

Abbreviations: DENV, dengue virus; SMD, standard mean difference; IQR, interquartile range; BMI, body mass index.

Chulaimbo) than at the coastal sites ($p<0.001$, Table 2). Consequently, DENV/malaria co-infection also was more frequent on the west. Comparing urban and rural areas, DENV/malaria co-infection appeared more frequent in rural sites (6% in rural vs 2.9% in urban settings, $p<0.001$).

Thousands of individually recorded symptoms and abnormal physical examination findings were manually reclassified into 113 clinical features. 48 clinical features that occurred at frequencies of five or less were considered sparse data and excluded from further analysis. Differences in the stratified frequencies of 28 of the remaining 65 clinical features were significant at the Bonferroni-adjusted $\alpha = 0.002$ (Table 3, italics, see S4 Table for additional supporting statistics). Fig 3 illustrates the frequencies of a selection of the most commonly encountered symptoms and physical exam findings, stratified by infection category.

Of the 2009 WHO clinical criteria for diagnosing probable dengue [26], vomiting was most frequently reported (28.6% of DENV solo-infected subjects). Yet, the odds of vomiting with DENV solo-infection were lower than those of malaria solo-infection ($OR_{malaria}$ 0.6, $CI_{99.8}$ 0.37 to 0.96, Table 4). Similarly, when compared to malaria solo-infected subjects, the odds of abdominal pain, a DENV warning sign, were lower for both DENV solo-infected ($OR_{malaria}$ 0.3, $CI_{99.8}$ 0.15 to 0.58) and DENV/malaria co-infected subjects ($OR_{malaria}$ 0.48, $CI_{99.8}$ 0.29 to 0.81). Only the criterion of rash appeared to be more likely in DENV solo-infected than malaria solo-infected subjects ($OR_{malaria}$ 3.9, $CI_{99.8}$ 1.-2 to 13). The odds of the remaining criteria for probable dengue, including nausea, aches and pains, lethargy, and hepatomegaly, were

**Table 3. Reported symptoms (^S) and physical exam observations (^O).**

| Clinical feature | DENV/malaria co-infection N = 265 (frequency %) | DENV solo-infection N = 220 (frequency %) | Malaria solo-infection N = 2880 (frequency %) | DENV/malaria uninfected N = 2843 (frequency %) | p |
|---|---|---|---|---|---|
| General | | | | | |
| Chills ^S | 58 (21.9) | 37 (16.8) | 792 (27.5) | 505 (17.8) | <0.001 |
| Malaise ^S | 104 (39.2) | 106 (48.2) | 1334 (46.3) | 1106 (38.9) | <0.001 |
| Aches and pains ^S | 69 (26.0) | 58 (26.4) | 953 (33.1) | 670 (23.6) | <0.001 |
| Loss of appetite ^S | 105 (39.6) | 98 (44.5) | 1057 (36.7) | 846 (29.8) | <0.001 |
| Itchiness ^S | 0 (0) | 1 (0.5) | 15 (0.5) | 9 (0.3) | 0.48 |
| Neurologic | | | | | |
| Irritable ^O | 0 (0) | 1 (0.5) | 4 (0.1) | 3 (0.1) | 0.41 |
| Headaches ^S | 130 (49.1) | 108 (49.1) | 1534 (53.3) | 1250 (44.0) | <0.001 |
| Dizziness ^S | 4 (1.5) | 6 (2.7) | 218 (7.6) | 95 (3.3) | <0.001 |
| Seizures ^S | 3 (1.1) | 3 (1.4) | 32 (1.1) | 15 (0.5) | 0.041 |
| Altered behavior ^S | 6 (2.3) | 0 (0) | 13 (0.5) | 5 (0.2) | <0.001 |
| Lethargic ^O | 71 (26.8) | 63 (28.6) | 637 (22.1) | 555 (19.5) | <0.001 |
| Neck stiffness ^S | 0 (0) | 0 (0) | 8 (0.3) | 3 (0.1) | 0.43 |
| Retro-orbital headache ^S | 1 (0.4) | 0 (0) | 10 (0.3) | 10 (0.4) | 1 |
| Gait abnormality ^O | 4 (1.5) | 0 (0) | 6 (0.2) | 2 (0.1) | 0.003 |
| Weakness ^O | 7 (2.6) | 5 (2.3) | 5 (0.2) | 7 (0.2) | <0.001 |
| Head and neck | | | | | |
| Coryza ^S | 0 (0) | 0 (0) | 17 (0.6) | 54 (1.9) | <0.001 |
| Red eyes ^S | 2 (0.8) | 8 (3.6) | 23 (0.8) | 33 (1.2) | 0.006 |
| Conjunctival injection ^O | 14 (5.3) | 22 (10.0) | 61 (2.1) | 70 (2.5) | <0.001 |
| Eye drainage ^S | 0 (0) | 1 (0.5) | 1 (0.0) | 7 (0.2) | 0.061 |
| Eye discharge ^O | 1 (0.4) | 4 (1.8) | 8 (0.3) | 21 (0.7) | 0.007 |
| Runny nose ^S | 70 (26.4) | 76 (34.5) | 601 (20.9) | 938 (33.0) | <0.001 |
| Nasal drainage ^O | 0 (0) | 6 (2.7) | 82 (2.8) | 81 (2.8) | 0.009 |
| Ear pain ^S | 1 (0.4) | 2 (0.9) | 15 (0.5) | 22 (0.8) | 0.54 |
| Sore throat ^S | 4 (1.5) | 5 (2.3) | 64 (2.2) | 178 (6.3) | <0.001 |
| Pharyngitis ^O | 2 (0.8) | 0 (0) | 53 (1.8) | 43 (1.5) | 0.096 |
| Mouth sores ^S | 0 (0) | 0 (0) | 4 (0.1) | 6 (0.2) | 0.80 |
| Altered taste ^S | 1 (0.4) | 0 (0) | 13 (0.5) | 13 (0.5) | 0.98 |
| Pulmonary | | | | | |
| Cough ^S | 115 (43.4) | 105 (47.7) | 996 (34.6) | 1337 (47.0) | <0.001 |
| Chest pain ^S | 1 (0.4) | 0 (0) | 5 (0.2) | 11 (0.4) | 0.37 |
| Difficulty breathing ^S | 3 (1.1) | 1 (0.5) | 17 (0.6) | 11 (0.4) | 0.26 |
| Dyspnea ^O | 1 (0.4) | 0 (0) | 2 (0.1) | 4 (0.1) | 0.36 |
| Tachypnea ^O | 7 (2.6) | 4 (1.8) | 65 (2.3) | 36 (1.3) | 0.02 |
| Wheezes ^O | 0 (0) | 0 (0) | 1 (0.0) | 11 (0.4) | 0.024 |
| Ronchi or rales ^O | 2 (0.8) | 1 (0.5) | 19 (0.7) | 53 (1.9) | <0.001 |
| Cardiac | | | | | |
| Tachycardia ^O | 35 (13.2) | 25 (11.4) | 407 (14.1) | 234 (8.2) | <0.001 |
| Delayed capillary refill ^O | 0 (0) | 0 (0) | 6 (0.2) | 3 (0.1) | 0.76 |
| Gastrointestinal | | | | | |
| Abdominal pain ^S | 46 (17.4) | 25 (11.4) | 872 (30.3) | 584 (20.5) | <0.001 |
| Abdominal tenderness ^O | 1 (0.4) | 2 (0.9) | 16 (0.6) | 8 (0.3) | 0.19 |
| Diffuse abd. tenderness ^O | 1 (0.4) | 1 (0.5) | 4 (0.1) | 0 (0) | 0.016 |

(*Continued*)

**Table 3.** (Continued)

| Clinical feature | DENV/malaria co-infection N = 265 (frequency %) | DENV solo-infection N = 220 (frequency %) | Malaria solo-infection N = 2880 (frequency %) | DENV/malaria uninfected N = 2843 (frequency %) | p |
|---|---|---|---|---|---|
| *Nausea* [S] | *20 (7.5)* | *11 (5.0)* | *232 (8.1)* | *146 (5.1)* | *<0.001* |
| *Vomiting* [S] | *102 (38.5)* | *63 (28.6)* | *1156 (40.1)* | *695 (24.4)* | *<0.001* |
| Diarrhea [S] | 25 (9.4) | 28 (12.7) | 273 (9.5) | 316 (11.1) | 0.12 |
| Bloody vomit or stools [S] | 0 (0) | 1 (0.5) | 5 (0.2) | 2 (0.1) | 0.26 |
| Hepatomegaly [O] | 0 (0) | 0 (0) | 6 (0.2) | 2 (0.1) | 0.63 |
| *Splenomegaly* [O] | *14 (5.3)* | *13 (5.9)* | *82 (2.8)* | *64 (2.3)* | *0.001* |
| Genitourinary | | | | | |
| Dysuria [S] | 1 (0.4) | 0 (0) | 1 (0) | 4 (0.1) | 0.18 |
| Musculoskeletal | | | | | |
| *Myalgia* [S] | *28 (10.6)* | *14 (6.4)* | *150 (5.2)* | *103 (3.6)* | *<0.001* |
| Joint pain [S] | 77 (29.1) | 85 (38.6) | 1001 (34.8) | 1030 (36.2) | 0.068 |
| Joint warmth [O] | 4 (1.5) | 2 (0.9) | 68 (2.4) | 54 (1.9) | 0.39 |
| *Joint erythema* [O] | *4 (1.5)* | *4 (1.8)* | *5 (0.2)* | *5 (0.2)* | *<0.001* |
| *Joint tenderness* [O] | *56 (21.1)* | *52 (23.6)* | *271 (9.4)* | *263 (9.3)* | *<0.001* |
| *Bones ache* [S] | *6 (2.3)* | *2 (0.9)* | *35 (1.2)* | *14 (0.5)* | *0.002* |
| Backaches [S] | 0 (0) | 0 (0) | 0 (0) | 7 (0.2) | 0.046 |
| Dermatologic | | | | | |
| *Rash* [S] | *4 (1.5)* | *8 (3.6)* | *53 (1.8)* | *94 (3.3)* | *0.002* |
| Rash [O] | 3 (1.1) | 9 (4.1) | 31 (1.1) | 48 (1.7) | 0.005 |
| *Maculopapular exanthem* [O] | *1 (0.4)* | *6 (2.7)* | *38 (1.3)* | *82 (2.9)* | *<0.001* |
| Skin or soft tissue infection [O] | 0 (0) | 2 (0.9) | 5 (0.2) | 4 (0.1) | 0.15 |
| Tinea corporis or capitis [O] | 0 (0) | 1 (0.5) | 15 (0.5) | 11 (0.4) | 0.70 |
| Scars [O] | 0 (0) | 0 (0) | 4 (0.1) | 2 (0.1) | 0.81 |
| Hematologic | | | | | |
| Pallor [O] | 0 (0) | 0 (0) | 15 (0.5) | 5 (0.2) | 0.12 |
| Bloody nose [S] | 2 (0.8) | 4 (1.8) | 11 (0.4) | 7 (0.2) | 0.011 |
| *Scleral icterus* [O] | *28 (10.6)* | *18 (8.2)* | *156 (5.4)* | *101 (3.6)* | *<0.001* |
| Lymphatic | | | | | |
| *Cervical adenopathy* [O] | *34 (12.8)* | *50 (22.7)* | *286 (9.9)* | *437 (15.4)* | *<0.001* |
| Axillary adenopathy [O] | 0 (0) | 5 (2.3) | 8 (0.3) | 15 (0.5) | 0.004 |
| Inguinal adenopathy [O] | 1 (0.4) | 0 (0) | 4 (0.1) | 12 (0.4) | 0.16 |

[S]: Reported symptom.

[O]: Observed physical exam finding.

Between-strata differences ($p \leq 0.002$, chi-square) are italicized (see S4 Table for additional statistical details). The following clinical variables occurred a total of 5 times or less, were considered sparse data and were excluded from the analysis: Lethargic [S], irritable [S], photophobia [S], serous otitis [O], left sided abdominal tenderness [O], epigastric tenderness [O], petechiae [O], bruising [O], bruising [S], hematuria [S], hyperreflexia [O], right sided abdominal tenderness [O], constipation [S], jaundice [O], swollen lymph nodes [S], facial swelling [O], lower abdominal tenderness [O], abdominal distension [O], skin hyperpigmentation [O], pallor [S], neck stiffness [O], hyporeflexia [O], itchy eyes [S], drainage from ear [S], mouth sores [O], thrush [O], bad breath [S], back tenderness [O], vesicles [O], insect bites [O], abrasion [O], jaundice [S], night sweats [S], postictal state [O], ataxia [O], sneezing [S], heart racing [S], heart murmur [O], absent bowel sounds [O], weakness [O], trauma [O], chest tenderness [O], facial rash [O], truncal rash [O], bleeding from gums [S], lower extremity edema [O], generalized edema [O], swelling of arms and legs [S].

Abbreviations: DENV, dengue virus; SMD, standard mean difference; IQR, interquartile range; BMI, body mass index.

no different between DENV or malaria infected individuals. Of note, WHO laboratory criteria for diagnosis of probable DENV, which include leukopenia and increased hematocrit, could not be tested routinely at clinic sites. The clinical officers also noted that the tourniquet test

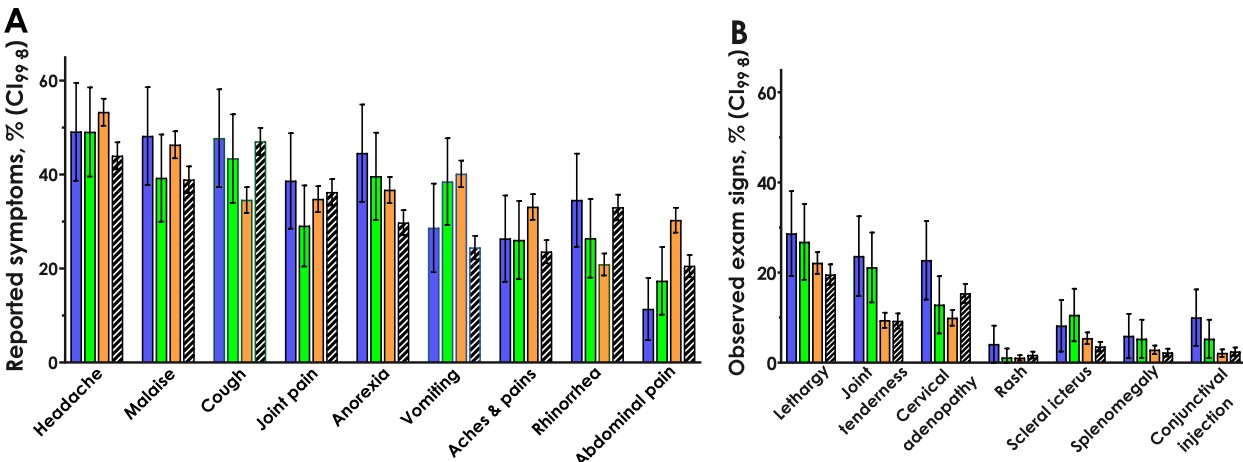

**Fig 3. Most frequently reported symptoms and observed physical exam findings.** Percent of subjects with symptoms (panel A) or abnormal physical examination findings (panel B) stratified by infection. DENV solo-infected, blue bars. DENV/malaria co-infected, green bars. Malaria solo-infected, orange bars. Subjects with neither DENV nor malaria, unshaded bars with slashes. Error bars represent exact binomial 99.8% confidence intervals.

was difficult to interpret, as petechiae were difficult to identify in the setting of dark skin. Taken together, these factors limited the utility of WHO diagnostic criteria for diagnosis of DENV in the clinical setting [14].

Use of the PedsQL™ 4.0 SF15 [18] HrQoL survey to assess infection-associated disability in the acute setting demonstrated high feasibility, with only 0.04% missing responses at the acute visit, and 0% missing at the convalescent visit. Regarding the utility of the survey, the ceiling effect (percent of subjects with maximum scores) for the overall score at the acute visit was 11.1%, suggesting the tool had sufficient resolution to detect differences in HrQoL. By the convalescent visit, 38.3% of subjects had maximum scores of 100, revealing a substantial ceiling effect, which made the survey less useful. Overall, the self- and parent-reported PedsQL scores appeared to be higher for DENV solo-infected and DENV/malaria co-infected subjects, compared to malaria solo-infected or DENV/malaria uninfected subjects (Fig 4).

There were no deaths associated with AFI encounters in this study. 249 (3.3% of 7509) children were recommended by the treating clinician to be admitted to hospital, of whom 217 (3.5% of 6208) were included in the complete case analysis. We observed no cases who met classification criteria for severe dengue based on symptoms and physical exam findings.

Malaria diagnosis as recorded by clinicians agreed with laboratory results in 81.8% of cases (Cohen's kappa = 0.64, $p<0.001$). The high prevalence of malaria (52.5%) contributed to the high positive and negative predictive values of clinician-diagnosed malaria (Table 5). In contrast, while DENV diagnosis as recorded by the clinical officer agreed with laboratory results in 79.8% of cases, the sensitivity for detecting DENV cases was poor (0.14, Table 6).

Prior to partitioning the complete cohort data into training (70%) and test (30%) sets, we imputed six missing age values using the R function missForest (OOB error 0.416 NRMSE). The overall prevalence of malaria infection in the test set was 50.7%. Clinicians outperformed all six models in identifying malaria (82.4% clinician accuracy compared with range of 53.4 to 68.7% for the models, Table 5). In contrast, the overall prevalence of DENV infection (7.8%) was over six-fold lower (Table 6). The clinicians accurately diagnosed DENV in 21 of 145 DENV-infected subjects, but also diagnosed DENV in 254 of the 1716 subjects that did not

**Table 4. Unadjusted odds ratios (OR$_{malaria}$) of clinical features by infection strata compared to malaria solo-infection as the reference (CI$_{99.8}$).**

| Clinical feature | DENV solo-infection | DENV/malaria co-infection | DENV/malaria uninfected |
|---|---|---|---|
| General | | | |
| Chills [S] | *0.53 (0.30 to 0.94)* | 0.74 (0.46 to 1.2) | *0.57 (0.47 to 0.69)* |
| Malaise [S] | 1.1 (0.7 to 1.7) | 0.75 (0.5 to 1.1) | *0.74 (0.63 to 0.87)* |
| Aches and pains [S] | 0.72 (0.44 to 1.2) | 0.71 (0.45 to 1.1) | *0.62 (0.52 to 0.75)* |
| Loss of appetite [S] | 1.4 (0.9 to 2.1) | 1.1 (0.75 to 1.7) | *0.73 (0.61 to 0.87)* |
| Itchiness [S] | 0.87 (0.004 to 21) | ~0 (~0 to ++) | 0.61 (0.16 to 2.2) |
| Neurologic | | | |
| Irritable [O] | ~0 (~0 to ~0) | ~0 (~0 to ~0) | ~0 (~0 to ~0) |
| Headaches [S] | 0.85 (0.55 to 1.3) | 0.84 (0.57 to 1.3) | *0.69 (0.58 to 0.81)* |
| Dizziness [S] | 0.34 (0.09 to 1.3) | *0.19 (0.04 to 0.90)* | *0.42 (0.29 to 0.62)* |
| Seizures [S] | 1.23 (0.19 to 8.1) | 1.0 (0.16 to 6.7) | 0.47 (0.18 to 1.2) |
| Altered behavior [S] | ~0 (~0 to ++) | 5.1 (0.1.1 to 22) | 0.39 (0.08 to 2.0) |
| Lethargic [O] | 1.4 (0.87 to 2.3) | 1.29 (0.82 to 2.0) | 0.85 (0.7 to 1.0) |
| Neck stiffness [S] | ~0 (~0 to ~0) | ~0 (~0 to ~0) | ~0 (~0 to ~0) |
| Retro-orbital headache [S] | ~0 (~0 to ~0) | ~0 (~0 to ~0) | ~0 (~0 to ~0) |
| Gait abnormality [O] | ~0 (~0 to ~0) | ~0 (~0 to ~0) | ~0 (~0 to ~0) |
| Weakness [O] | *13.4 (1.9 to 96)* | *15.6 (2.5 to 97)* | 1.4 (0.23 to 8.7) |
| Head and neck | | | |
| Coryza [S] | ~0 (~0 to ++) | ~0 (~0 to ~0) | *3.3 (1.4 to 7.73)* |
| Red eyes [S] | *4.7 (1.3 to 17)* | 0.94 (0.01 to 9.3) | 1.5 (0.63 to 3.4) |
| Conjunctival injection [O] | *5.1 (2.3 to 11)* | 2.6 (1 to 6.7) | 1.2 (0.67 to 2.0) |
| Eye drainage [S] | ~0 (~0 to ~0) | ~0 (~0 to ~0) | ~0 (~0 to ~0) |
| Eye discharge [O] | 6.6 (0.99 to 45) | 1.4 (0.051 to 36) | 2.7 (0.74 to 9.7) |
| Runny nose [S] | *2 (1.3 to 3.2)* | 1.4 (0.87 to 2.1) | *1.9 (1.5 to 2.3)* |
| Nasal drainage [O] | 0.96 (0.25 to 3.6) | ~0 (~0 to ~0) | 1 (0.61 to 1.6) |
| Ear pain [S] | 1.8 (0.17 to 18) | 0.72 (0.03 to 18) | 1.5 (0.53 to 4.2) |
| Sore throat [S] | 1.0 (0.24 to 4.4) | 0.67 (0.14 to 3.4) | *2.9 (1.9 to 4.6)* |
| Pharyngitis [O] | ~0 (~0 to ~0) | 0.41 (0.043 to 3.8) | 0.82 (0.43 to 1.6) |
| Mouth sores [S] | ~0 (~0 to ~0) | ~0 (~0 to ~0) | ~0 (~0 to ~0) |
| Altered taste [S] | ~0 (~0 to ++) | 0.84 (0.034 to 21) | 1.0 (0.30 to 3.4) |
| Pulmonary | | | |
| Cough [S] | *1.7 (1.1 to 2.7)* | 1.5 (0.97 to 2.2) | *1.7 (1.4 to 2.0* |
| C)hest pain [S] | ~0 (~0 to ~0) | ~0 (~0 to ~0) | ~0 (~0 to ~0) |
| Difficulty breathing [S] | 0.77 (0.032 to 19) | 1.9 (0.28 to 13) | 0.65 (0.20 to 2.2) |
| Dyspnea [O] | ~0 (~0 to ~0) | ~0 (~0 to ~0) | ~0 (~0 to ~0) |
| Tachypnea [O] | 0.80 (0.16 to 4.0) | 1.2 (0.34 to 4.1) | 0.56 (0.29 to 1.1) |
| Wheezes [O] | ~0 (~0 to ~0) | ~0 (~0 to ~0) | ~0 (~0 to ~0)) |
| Ronchi or rales [O] | 0.69 (0.029 to 16) | 1.2 (0.11 to 11) | *2.9 (1.2 to 6.6)* |
| Cardiac | | | |
| Tachycardia [O] | 0.78 (0.40 to 1.5) | 0.92 (0.52 to 1.7) | *0.54 (0.42 to 0.71)* |
| Delayed capillary refill [O] | ~0 (~0 to ~0) | ~0 (~0 to ~0) | ~0 (~0 to ~0) |
| Gastrointestinal | | | |
| Abdominal pain [S] | *0.30 (0.15 to 0.58)* | *0.48 (0.29 to 0.81)* | *0.60 (0.49 to 0.72)* |
| Abdominal tenderness [O] | 1.6 (0.16 to 17) | 0.68 (0.028 to 16) | 0.51 (0.13 to 1.9) |
| Diffuse abd. tenderness [O] | ~0 (~0 to ~0) | ~0 (~0 to ~0)17) | ~0 (~0 to ~0) |
| Nausea [S] | 0.60 (0.23 to 1.6) | 0.93 (0.44 to 2.0) | *0.62 (0.44 to 0.87)* |

*(Continued)*

**Table 4.** (Continued)

| Clinical feature | DENV solo-infection | DENV/malaria co-infection | DENV/malaria uninfected |
|---|---|---|---|
| Vomiting [S] | *0.60 (0.37 to 0.96)* | 0.93 (0.62 to 1.4) | *0.48 (0.40 to 0.58)* |
| Diarrhea [S] | 1.4 (0.72 to 2.7) | 0.99 (0.50 to 2.0) | 1.2 (0.91 to 1.6) |
| Bloody vomit or stools [S] | ~0 (~0 to ~0) | ~0 (~0 to ~0) | ~0 (~0 to ~0) |
| Hepatomegaly [O] | ~0 (~0 to ~0) | ~0 (~0 to ~0) | ~0 (~0 to ~0) |
| Splenomegaly [O] | 2.1 (0.83 to 5.5) | 1.9 (0.76 to 4.8) | 0.79 (0.47 to 1.3) |
| Genitourinary | | | |
| Dysuria [S] | ~0 (~0 to ~0) | ~0 (~0 to ~0) | ~0 (~0 to ~0) |
| Musculoskeletal | | | |
| Myalgia [S] | 1.2 (0.51 to 3.0) | *2.2 (1.1 to 4.2)* | 0.68 (0.46 to 1.0) |
| Joint pain [S] | 1.2 (0.76 to 1.8) | 0.77 (0.50 to 1.2) | 1.1 (0.90 to 1.3) |
| Joint warmth [O] | 0.38 (0.041 to 3.5) | 0.63 (0.13 to 3.1) | 0.8 (0.45 to 1.4) |
| Joint erythema [O] | ~0 (~0 to ~0) | ~0 (~0 to ~0) | ~0 (~0 to ~0) |
| Joint tenderness [O] | *3.0 (1.8 to 5.1)* | *2.6 (1.6 to 4.3)* | 0.98 (0.74 to 1.3) |
| Bones ache [S] | 0.75 (0.078 to 7.1) | 1.9 (0.47 to 7.5) | 0.4 (0.15 to 1.1) |
| Backaches [S] | ~0 (~0 to ~0) | ~0 (~0 to ~0) | ~0 (~0 to ~0) |
| Dermatologic | | | |
| Rash [S] | 2.0 (0.61 to 6.6) | 0.82 (0.16 to 4.1) | *1.8 (1.1 to 3.1)* |
| Rash [O] | *3.9 (1.2 to 13)* | 1.1 (0.16 to 6.9) | 1.6 (0.77 to 3.3) |
| Maculopapular exanthem [O] | 2.1 (0.53 to 8.3) | 0.28 (0.012 to 6.5) | *2.2 (1.2 to 4.1)* |
| Skin or soft tissue infection [O] | ~0 (~0 to ~0) | ~0 (~0 to ~0) | ~0 (~0 to ~0) |
| Tinea corporis or capitis [O] | 0.87 (0 to 8.7) | ~0 (~0 to ~0) | 0.74 (0.22 to 2.5) |
| Scars [O] | ~0 (~0 to ~0) | ~0 (~0 to ~0) | ~0 (~0 to ~0) |
| Hematologic | | | |
| Pallor [O] | ~0 (~0 to ~0) | ~0 (~0 to ~0) | ~0 (~0 to ~0) |
| Blood nose [S] | 4.8 (0.78 to 30) | 2.0 (0.18 to 22) | 0.64 (0.14 to 2.9) |
| Scleral icterus [O] | 1.6 (0.70 to 3.5) | *2.1 (1.1 to 4)* | *0.64 (0.43 to 0.96)* |
| Lymphatic | | | |
| Cervical adenopathy [O] | *2.7 (1.6 to 4.5)* | 1.3 (0.73 to 2.4) | *1.6 (1.3 to 2.1)* |
| Axillary adenopathy [O] | *8.3 (1.1 to 49)* | 0.0013 (~0 to ++) | 1.9 (0.49 to 7.4) |
| Inguinal adenopathy [O] | ~0 (~0 to ~0) | ~0 (~0 to ~0) | ~0 (~0 to ~0) |

[S] Reported as a symptom.

[O] Observed during physical examination.

ORs were not calculated for features that had sparse frequencies (n$\leq$5).

For ease of viewing, values $>10^4$ are denoted as "++" and values $<10^{-4}$ are denoted as "~0".

$p\leq$0.002 are italicized.

have DENV. Of the machine learning models, boosted logistic regression was the only model that identified any DENV cases (8 of 145). Neither the clinicians nor any of the models identified any of the 79 cases (4.2% prevalence) of DENV/malaria co-infection in the test dataset. Of note, clinicians accurately excluded both DENV and malaria 71% of the time. Prediction of absence of both DENV and malaria by the machine learning models ranged in accuracy from 46 to 61% (S5 Table).

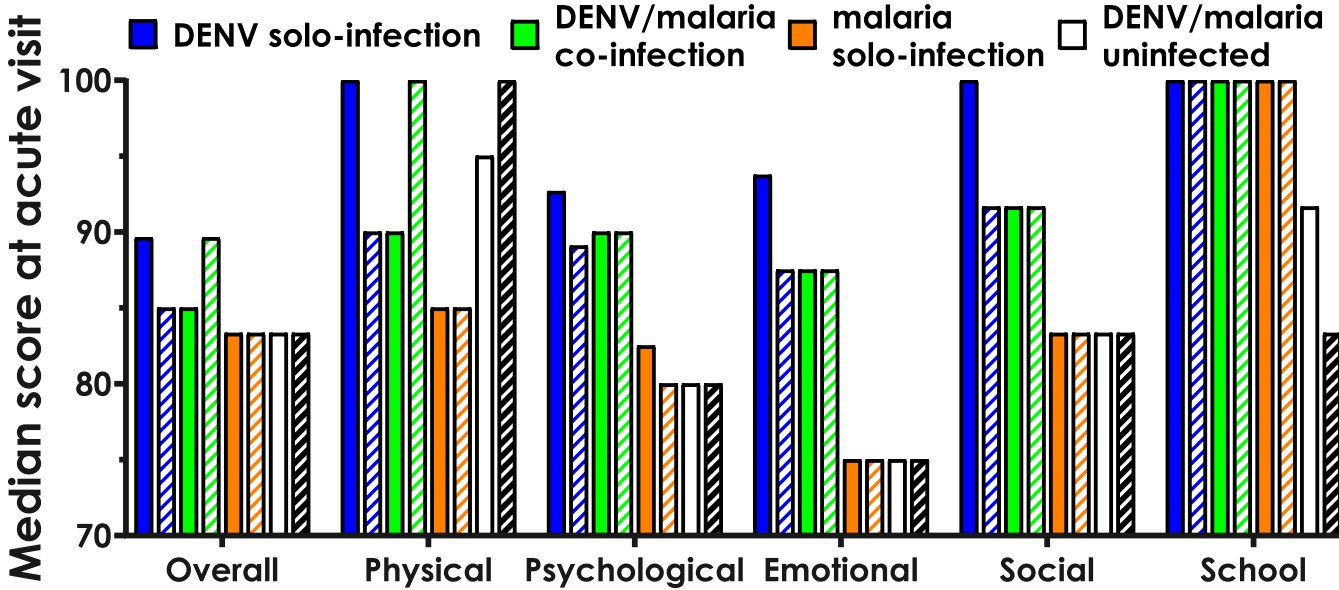

**Fig 4. PedsQL scores at time of febrile visit stratified by infection.** Open bars, subject self-reported scores. Slashed bars, parent-reported proxy scores. DENV solo-infected, blue. DENV/malaria co-infected, green. Malaria solo-infected, orange. Subjects with neither DENV nor malaria, unshaded. $p \leq 0.002$ for all between-strata comparisons by Kruskal-Wallis H test.

## Discussion

The goals of this study were to describe the symptoms and physical exam findings of DENV and malaria infections in Kenyan children, and to investigate whether differences observed in such clinical manifestations are sufficient for the identification of DENV infection in absence of diagnostic testing in an area of high malaria prevalence. Out of 65 clinical features that were observed at a frequency of 6 or more, differences in frequencies of 28 clinical features could be detected between DENV or malaria solo-infected, DENV/malaria co-infected, and DENV/malaria uninfected children at $p \leq 0.002$ (Table 3). However, some features considered cardinal signs of DENV infection, including vomiting and abdominal pain, were observed more frequently in malaria infected children than in DENV infected children. Thus, the overlapping symptomatology of DENV and malaria infections impaired the utility of the WHO criteria for diagnosis of probable DENV infection in our cohort. This may have contributed to why

**Table 5. Performance characteristics for predicting malaria infection in validation dataset.**

| Laboratory diagnosis (n = 1861) | TP (n = 943) | FN | FP | TN (n = 918) | Acc | Sens | Spec | PPV | NPV | Kappa | Tuning parameter |
|---|---|---|---|---|---|---|---|---|---|---|---|
| Clinician diagnosis | 803 | 140 | 188 | 730 | 0.82 | 0.85 | 0.80 | 0.81 | 0.84 | 0.64 | NA |
| Boosted logistic regression | 332 | 661 | 185 | 733 | 0.57 | 0.35 | 0.80 | 0.64 | 0.55 | 0.15 | nIter = 11 |
| Classification tree | 385 | 558 | 224 | 694 | 0.58 | 0.41 | 0.76 | 0.63 | 0.55 | 0.16 | Cp = 0.0177 |
| Random forest | 608 | 335 | 283 | 635 | 0.67 | 0.64 | 0.66 | 0.68 | 0.65 | 0.34 | Mtry = 72 |
| Support vector machines | 595 | 348 | 234 | 684 | 0.69 | 0.63 | 0.75 | 0.72 | 0.66 | 0.38 | Sigma = 0.0324, C = 1 |
| Naïve Bayes | 889 | 54 | 813 | 105 | 0.53 | 0.94 | 0.11 | 0.52 | 0.66 | 0.058 | Laplace = 0, Usekernel = F, Adjust = 1 |
| Neural networks (MLP) | 749 | 194 | 474 | 444 | 0.64 | 0.79 | 0.48 | 0.61 | 0.70 | 0.28 | Size = 3, decay = 0 |

Abbreviations: TP, true positive; FN, false negative; FP, false positive; TN, true negative; Acc, accuracy; Sens, sensitivity; Spec, specificity; PPV, positive predictive value; NPV, negative predictive value; MLP, multi-layer perceptron; nIter, number of iterations; Cp, complexity parameter; Mtry, number of trees.

**Table 6. Performance characteristics for predicting DENV infection in validation dataset.**

| Laboratory diagnosis (n = 1861) | TP (n = 145) | FN | FP | TN (1716) | Acc | Sens | Spec | PPV | NPV | Kappa | Tuning parameter |
|---|---|---|---|---|---|---|---|---|---|---|---|
| Clinician diagnosis | 21 | 124 | 254 | 1462 | 0.80 | 0.14 | 0.85 | 0.076 | 0.92 | -0.0023 | NA |
| Boosted logistic regression | 8 | 137 | 29 | 1687 | 0.91 | 0.055 | 0.98 | 0.22 | 0.92 | 0.058 | nIter = 11 |
| Classification tree | 0 | 145 | 0 | 1716 | 0.92 | 0 | 1 | NA | 0.92 | 0 | Cp = 0.0177 |
| Random forest | 0 | 145 | 0 | 1716 | 0.92 | 0 | 1 | NA | 0.92 | 0 | Mtry = 72 |
| Support vector machine | 0 | 145 | 0 | 1716 | 0.92 | 0 | 1 | NA | 0.92 | 0 | Sigma = 0.0324, C = 1 |
| Naïve Bayes | 0 | 145 | 0 | 1716 | 0.92 | 0 | 1 | NA | 0.92 | 0 | Laplace = 0, Usekernel = F, Adjust = 1 |
| Neural networks (MLP) | 0 | 145 | 0 | 1716 | 0.92 | 0 | 1 | NA | 0.92 | 0 | Size = 3, decay = 0 |

Abbreviations: TP, true positive; FN, false negative; FP, false positive; TN, true negative; Acc, accuracy; Sens, sensitivity; Spec, specificity; PPV, positive predictive value; NPV, negative predictive value; MLP, multi-layer perceptron; nIter, number of iterations; Cp, complexity parameter; Mtry, number of trees.

clinician sensitivity for diagnosis of malaria was high (85%), whereas sensitivity for diagnosis of DENV was low (14%). Leveraging our dataset of over six thousand complete cases, we investigated whether artificial intelligence could identify and distinguish DENV from malaria infections using the clinical features available to the clinicians. The naïve Bayes machine learning model demonstrated a higher sensitivity for detecting malaria compared with clinicians (94 vs 84%, respectively), but carried a lower specificity (11 vs 80%, respectively). Clinician diagnosis of malaria outperformed the remaining machine learning models regarding accuracy, sensitivity, and specificity.

It is important to note, however, that clinicians had access to the malaria testing that was available at their clinic at the time of the visit to assist them in their diagnoses, while laboratory testing data were withheld from machine learning algorithms. DENV diagnostic performance was poor for both clinicians and artificial intelligence. In fact, boosted logistic regression was the only algorithm to classify any DENV case. Thus, neither human nor machine could reliably detect a DENV prevalence of up to 8%.

The PedsQL survey was designed to assess the disability burden of chronic disease on functions of daily life in children [18], so our use of the tool in the setting of acute infection was exploratory. We detected differences in scores at the acute visit, suggesting this may be a useful tool to uncover otherwise hidden burdens of infectious diseases; however, we did not detect between-strata differences at the convalescent visit that might suggest long-term effects of DENV infection.

Our study was intended to use only the data that are reliably available to clinicians (symptom recollection and physical exam findings) but was limited by some technical considerations. We collected vital signs, including heart rate, respiratory rate, and blood pressure, but excluded these from the analysis due to variations in the data that raised concerns for consistency and accuracy of the measurements. Many subjects were lost to follow up, which decreased the number of subjects selected for the complete cases analysis. Our analysis of differences in clinical manifestations between infection strata may be susceptible to false discovery due to multiple comparisons. It is important to note that $p < 0.001$ for most of the observed between-strata differences in symptom frequency (Table 3). Bias in clinician education, training, and experience can limit the generalizability of our assessment of clinician-based diagnostic performance. We did not perform dimension reduction [27] prior to developing the machine learning models, as most features were binary. Another important limitation of our data is our study collected data on hospitalization as a clinical outcome, which we had considered a proxy for disease severity. However, we learned that the decision to hospitalize was

influenced by the ability of the child's guardians to pay and the availability of hospital beds. While these may have contributed to underestimation of hospitalization rates, remarkably, there were no deaths recorded among study participants.

Our detection of DENV cases during most months over the five-year duration of our study, including in between outbreak periods (Fig 2), suggests that endemic transmission of DENV also may be ongoing in Kenya [28]. This finding has important implications for public health policy. For example, approaches to interrupt endemic transmission are different from those used to target outbreaks. The observed DENV/malaria co-infections further heighten the concern that, in the absence of routine testing for DENV, malaria-attributed morbidity and mortality in Kenya may be due to misdiagnosed co-infections. Such misclassification can have widespread impact on public health decision-making. For instance, resource allocation decisions for vector management vary with the different behaviors of DENV transmitting *Aedes* mosquitoes and of malaria transmitting *Anopheles* mosquitoes. In resource restricted settings, inaccurate epidemiologic data can misdirect the use of limited resources for public health interventions.

The need for more accessible testing for emerging and re-emerging infections such as DENV is widely recognized but inconsistently prioritized. In the absence of testing, detection of emerging infectious outbreaks relies on clinician observation and suspicion. Our study evaluated the real-world application of clinical assessment, using the only reliably available tools to the clinician: their observations. Clinicians performed well diagnosing malaria (sensitivity 85.1%) in the setting of high prevalence (50.2%). The prevalence of DENV infection during our study (7.9%) was lower than that of malaria but was nevertheless high enough to represent transmission at epidemic proportions. Yet, clinicians who had been educated on the signs and symptoms of DENV infection prior to the study, were unable to reliably detect DENV infection (sensitivity 14.5%). Thus, without laboratory testing, passive clinical reporting of DENV infection is not a reliable means for public health surveillance for this re-emerging infection. A similar study to ours uncovered a large burden of chikungunya virus infection among Kenyan children [29]. The high prevalence and force of infection of malaria, in combination with lack of routine surveillance for arboviruses such as DENV or chikungunya virus, contribute towards under-recognition of alternate or concomitant causes for AFI and over-diagnosis of malaria [30].

## Conclusion

International coordination is critical for pandemic preparedness. Yet, our study highlights inequitable access to diagnostic testing as a fundamental flaw in global surveillance for DENV, which we present as a model for other emerging infections. Improving diagnostic capability, particularly in resource-restricted areas of the world, should be the top priority for all nations, lest our ability to detect emerging infections continue to be severely hampered. As epidemics and pandemics have demonstrated time and again, transmission of infections does not respect international boundaries.

The large number of DENV/malaria co-infections identified should alert clinicians in areas of high malaria transmission that co-infection should be considered in the evaluation of acute febrile illness, even when a diagnosis of malaria has been established. The principle of diagnostic parsimony cannot be universally applied in the practice of medicine. Rather, local epidemiologic data must be carefully considered in order to achieve accurate diagnoses and appropriate clinical care.

## Supporting information

**S1 Checklist. STROBE Statement—Checklist of items that should be included in reports of** *cohort studies.*
(DOCX)

**S1 Table. Characteristics of all enrolled subjects with acute febrile illness by region or urban/rural.**
(DOCX)

**S2 Table. Characteristics of all enrolled and complete cases cohort.**
(DOCX)

**S3 Table. Supporting statistics for Table 2.**
(DOCX)

**S4 Table. Supporting statistics for Table 3.**
(DOCX)

**S5 Table. Performance characteristics for predicting absence of DENV and absence of malaria infection in the validation dataset.**
(DOCX)

**S1 Text. Demographic survey.**
(DOCX)

**S2 Text. PedsQL surveys translated into Kiswahili and Dhuluo.**
(PDF)

**S3 Text. PLoS inclusivity in global research checklist.**
(PDF)

## Acknowledgments

The Stanford REDCap platform (http://redcap.stanford.edu) is developed and operated by Stanford Medicine Research IT team. The REDCap platform services at Stanford are subsidized by a) Stanford School of Medicine Research Office, and b) the National Center for Research Resources and the National Center for Advancing Translational Sciences, National Institutes of Health, through grant KL2 TR001085.

## Author Contributions

**Conceptualization:** David M. Vu, Bryson A. Ndenga, Francis M. Mutuku, Dunstan Mukoko, Charles H. King, A. Desiree LaBeaud.

**Data curation:** David M. Vu, Amy R. Krystosik, Bryson A. Ndenga, Francis M. Mutuku, Carren M. Bosire, Claire Heath, Said Lipi Malumbo, Jael Sagina Amugongo.

**Formal analysis:** David M. Vu, Amy R. Krystosik.

**Funding acquisition:** David M. Vu, A. Desiree LaBeaud.

**Investigation:** David M. Vu, Bryson A. Ndenga, Francis M. Mutuku, Kelsey Ripp, Elizabeth Liu, Carren M. Bosire, Claire Heath, Philip Chebii, Priscilla Watiri Maina, Zainab Jembe, Said Lipi Malumbo, Jael Sagina Amugongo, Charles Ronga, Victoria Okuta, Noah Mutai, Nzaro G. Makenzi, Kennedy A. Litunda.

**Methodology:** David M. Vu, Carren M. Bosire, Claire Heath, A. Desiree LaBeaud.

**Project administration:** Bryson A. Ndenga, Francis M. Mutuku.

**Supervision:** Bryson A. Ndenga, Francis M. Mutuku, Carren M. Bosire, Dunstan Mukoko, Charles H. King, A. Desiree LaBeaud.

**Validation:** David M. Vu.

**Writing – original draft:** David M. Vu.

**Writing – review & editing:** David M. Vu, Amy R. Krystosik, Bryson A. Ndenga, Francis M. Mutuku, Kelsey Ripp, Elizabeth Liu, Carren M. Bosire, Claire Heath, Philip Chebii, Priscilla Watiri Maina, Zainab Jembe, Said Lipi Malumbo, Jael Sagina Amugongo, Charles Ronga, Victoria Okuta, Noah Mutai, Nzaro G. Makenzi, Kennedy A. Litunda, Dunstan Mukoko, Charles H. King, A. Desiree LaBeaud.

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
