## [Decision Letter · Decision Letter 0]

7 Mar 2023

PGPH-D-22-02073

Human or machine? Detection of acute dengue virus infection, with and without concurrent malaria infection, at Kenyan clinics in a cohort of acutely febrile children, 2014-2019

Dear Dr. Vu,

Thank you for submitting your manuscript to PLOS Global Public Health. After careful consideration, we feel that it has merit but does not fully meet PLOS Global Public Health’s publication criteria as it currently stands. Therefore, we invite you to submit a revised version of the manuscript that addresses the points raised during the review process.

EDITOR: Please insert comments here and delete this placeholder text when finished. Be sure to:

Indicate which changes you require for acceptance versus which changes you recommendAddress any conflicts between the reviews so that it's clear which advice the authors should followProvide specific feedback from your evaluation of the manuscript

Please ensure that your decision is justified on PLOS Global Public Health’s publication criteria and not, for example, on novelty or perceived impact.

We look forward to receiving your revised manuscript.

Kind regards,

Mathieu Nacher

Academic Editor

Journal Requirements:

2. Our staff editors have determined that your manuscript is likely within the scope of our Diagnostics in Global Health Call for Papers. This editorial initiative is headed by a team of Guest Editors for PLOS GPH: Senjuti Saha (Child Health Research Foundation, Bangladesh) and Titus Divala (Public Health Scotland, University of Glasgow and University of Malawi College of Medicine). The Collection will encompass a diverse range of research articles about diagnostics in global health, including innovation and deployment of point of care diagnostics; subsets of diagnostics related to infectious diseases, chronic diseases and injuries; policies related to and regulation of diagnostics; supply chain issues; and the affordability, accessibility, and availability of essential diagnostics.  Additional information can be found on our announcement page: https://collections.plos.org/call-for-papers/diagnostics-in-global-health/

If you would like your manuscript to be considered for this collection, please let us know in your cover letter and we will ensure that your paper is treated as if you were responding to this call.  Please note that being considered for the Collection does not require additional peer review beyond the journal’s standard process and will not delay the publication of your manuscript if it is accepted by PLOS GPH. If you would prefer to remove your manuscript from collection consideration, please specify this in the cover letter.

3. Please provide separate figure files in .tif or .eps format for Fig 2.

4. In the online submission form, you indicated that your data will be submitted to the Dryad database upon acceptance. Should your submission be accepted, we will require the following information in your Data Availability Statement: 

a. The DOI provided by Dryad

b. The citation for your data package in the reference section of your manuscript

c. The citation for your data package in the methods section

If you are unable to adhere to our open data policy, please kindly revise your statement to explain your reasoning and we will seek the editor's input on an exemption. Please be assured that, once you have provided your new statement, the assessment of your exemption will not hold up the peer review process.

Additional Editor Comments (if provided):

Reviewers' comments:

Reviewer's Responses to Questions

**Comments to the Author**

1. Does this manuscript meet PLOS Global Public Health’s publication criteria? Is the manuscript technically sound, and do the data support the conclusions? The manuscript must describe methodologically and ethically rigorous research with conclusions that are appropriately drawn based on the data presented.

Reviewer #1: Partly

Reviewer #2: Yes

2. Has the statistical analysis been performed appropriately and rigorously?

Reviewer #1: Yes

Reviewer #2: Yes

3. Have the authors made all data underlying the findings in their manuscript fully available (please refer to the Data Availability Statement at the start of the manuscript PDF file)?

Reviewer #1: Yes

Reviewer #2: No

4. Is the manuscript presented in an intelligible fashion and written in standard English?

Reviewer #1: Yes

Reviewer #2: Yes

5. Review Comments to the Author

Reviewer #1: This is an interesting piece of work on two important public health issues, malaria and dengue fever, which health systems in low- and middle-income countries are trying to address with little or no resources compared to wealthier countries. Nevertheless, this work carried out in kenya raises the following comments:

The presentation of the origin of the data, the organization and the flow of the data should be presented more early and more clearly in the text. The baseline values should appear directly in the text and the differences observed between the study sites should be more discussed. Thus, by further clarifying the context and origin of the data, it will be clearer to understand the specific organization and specific goals of the analysis.

In general, the authors do not sufficiently present the statistical tools used and/or the statistical results obtained. For example, concerning the clinical manifestations carried out on several categories, when they are significant, are not sufficiently informative: they do not show which value carries the weight of the significant difference (e.g. the contribution of Chi2). In the same way, the results of these tests are insufficiently discussed in the text.

The same remark can be made for the use of the 'misforest' package of R. The authors do not discuss the configurations used and do not present sufficiently the different means used (boosted logistic regression, classification tree, random forest, support vector machines, naive Bayes, neural netork) to evaluate the prediction performances of malaria and dengue. For a reader not familiar with this type of analysis, which I am, it is difficult to judge the interest or performance of each of them. Thus, without understanding the specific interest of each approach, it is difficult or impossible to appreciate the comparison of their results and to reach the same conclusion as the authors.

Reviewer #2: 1

Manuscript Number: PGPHU-D-22-02073

Title: Human or machine? Detection of acute dengue virus infection, with and without concurrent

malaria infection, at Kenyan clinics in a cohort of acutely febrile children, 2014-2019

Article Type: Research article

General comment

The differential diagnosis between child malaria and pediatric forms of arboviral diseases in areas

where both types of pathogens cocirculate may be challenging, especially in regions where there are

an overwhelming prominence of malaria and where clinicians are not familiar with arboviral diseases.

In addition, coinfection of malaria with arboviruses often purports severe disease which is difficult to

treat in the absence of readily available diagnostic tools like rapid diagnostic tests and point-of-care

facilities. For this purpose, researchers have developed clinical prediction models aimed at substituting

deficient lab facilities with diagnostic tools offering sufficient diagnostic accuracy (i.e., the best

compromise between sensitivity and specificity) for triaging and prompt management of patients.

In this framework, David Vu et al., report five years of surveillance of child acute febrile illness (AFI) in

four Kenyan clinics (two urban and two rural, two on coastal and two in inner-country Kenya) in the

aim to assess the diagnostic performance of clinical diagnosis, as compared to several machine learning

driven models for discriminating dengue, coinfection of dengue with malaria, or absence of both

conditions from malaria alone(reference). The paper is interesting and well-written. The findings

should have important implications for clinical management of AFIs in malaria and dengue endemic

countries. Prior to publication, the paper should benefit of some clarifications and minimal revision

including for the most important, additional details on machine learning and deep learning methods,

better consideration of STROBE recommendations for reporting the findings of observational research.

Optional revision could consist in performing a multinomial logistic regression model aimed at

distinguishing the factors associated with each infection stratum, namely dengue alone, or with

dengue with malaria co-infection, or absence of both conditions, taking malaria alone as referent

stratum (justified by the report that malaria is by far the predominant condition). If the goal is really

to distinguish each stratum, a multinomial logistic regression model is the best way to do, which

minimize the alpha risk, whereas separate logistic regression models carry the risk of identifying

predictors that are not associated when all strata are accounted simultaneously (as in real life).

Specific comments

Title and abstract

1(a). Done. No claim

1(b). The title is too long. I would remove “Human or machine” which is not clear and add “using

machine learning” after infection.

Page 1, line 44. “To train” applies for machine learning. Maybe prefer “develop” better appropriate

for clinical prediction models.

Page 1, line 46. “To test” applies for machine learning. Maybe prefer “validate” better appropriate

for clinical prediction models.

Page 1, lines 48 to 50. “In the Reserved test data” is not explicit. Prefer “In the validation dataset”.

Provide the same indicators for each model (accuracy, sensitivity and specificity)

2

Introduction

2. Background/rationale

Explain the scientific background and rationale for the investigation being reported.

Page 2, line 60 to page 3, line 61. Though the principle of parsimony may be known of a few, it is a

philosophical concept that deserves to be explained in a few words for a large majority (e.g, otherwise

said...).

Page 3, line 69. The term “rubrics” seems informatic jargon. Explain or replace by more explicit term.

3. Objectives

State specific objectives, including any prespecified hypotheses. Hypotheses are lacking.

Methods

4. Study design

Present key elements of study design early in the paper. Done. No claim.

5. Setting

Describe the setting, locations, and relevant dates, including periods of recruitment, exposure,

follow-up, and data collection.

Page 4, line 81. Be more precise on dates and specify if possible days.

6. Participants

6 (a). Cohort study—Give the eligibility criteria, and the sources and methods of selection of

participants. Describe methods of follow-up

Page 4, lines 107 to 109. Dengue diagnosis relies on RT-PCR (gold standard) and IgG antibody

seroconversion. Why not using rapid diagnostic tests as recommended by the WHO ?

7. Variables

Clearly define all outcomes, exposures, predictors, potential confounders, and effect modifiers. Give

diagnostic criteria, if applicable

8. Data sources/ measurement

For each variable of interest, give sources of data and details of methods of assessment

(measurement). Describe comparability of assessment methods if there is more than one group

9. Bias

Describe any efforts to address potential sources of bias

10. Study size

Explain how the study size was arrived at. Not appropriate. No claim.

3

11. Quantitative variables

Explain how quantitative variables were handled in the analyses. If applicable, describe which

groupings were chosen and why

12. Statistical methods

12 (a). Describe all statistical methods, including those used to control for confounding

Page 5, lines 116 to 117. Write “Kruskal-Wallis” and not “Kruskall-Wallis”. Simplify the statement and

write “Non parametric Kruskal-Wallis H test and Student T test for continuous data, as appropriate.”

Page 5, lines 119 to 120. I don’t understand the justification of setting the Bonferroni correction to

alpha=0.002 given each variable is compared three times (dengue alone vs malaria alone, denguemalaria coinfection vs malaria alone, no dengue no malaria vs malaria alone).

Page 5, lines 123 to 125. Detail machine learning algorithms in a few words with their particularities,

applications and limits.

Page 5, line 137. Replace “To train the models” by to “to develop the models.” Replace “was reserved

to test” by “was validated”

12 (b). Describe any methods used to examine subgroups and interactions

12 (c). Explain how missing data were addressed

12 (d). Cohort study—If applicable, explain how loss to follow-up was addressed

12 (e). Describe any sensitivity analyses

Results

13. Participants

13 (a). Report numbers of individuals at each stage of study—eg numbers potentially eligible,

examined for eligibility, confirmed eligible, included in the study, completing follow-up, and analyse

13 (b). Done. No claim.

13 (c). Done. No claim.

14. Descriptive data

14 (a). Give characteristics of study participants (eg demographic, clinical, social) and information on

exposures and potential confounders

Provide average delays of presentation to hospital since days of symptom onset for each stratum.

Page 7, Table 1. Write “Kruskal-Wallis” and not “Kruskall-Wallis”.

Page 7, line 156. “Differences in HAZ and BAZ” between which groups ?

Page 8, Table 2. Write “Kruskal-Wallis” and not “Kruskall-Wallis”.

4

14 (b). Indicate number of participants with missing data for each variable of interest

14 (c). Cohort study—Summarise follow-up time (eg, average and total amount)

If possible, specify the attrition for each group at time of visit (one month).

15. Outcome data

15 (a). Cohort study—Report numbers of outcome events or summary measures over time

Done in table 3. No claim.

Page 10, line 172. Among the 65 variables, there are 23 variables significant at alpha=0.002 and not

28

Page 10, lines 172 to 174. It is unclear how the variables presented in Fig 3 were selected. Explain.

Figure 3 is heavy loaded and most credibility intervals overlap. I would simplify and use a Venn

Diagram in which I would position each symptom/sign.

16. Main results

16 (a). Give unadjusted estimates and, if applicable, confounder-adjusted estimates and their

precision (eg, 95% confidence interval). Make clear which confounders were adjusted for and why

they were included

Specify in table 4 that OR are crude (unadjusted). If the purpose here is to distinguish the symptoms

and clinical signs characteristic of each stratum, why not have preferred a multinomial logistic

regression model, instead of performing three models against malaria alone as referent category.

These should be corrected for alpha risk inflation, whereas a multinomial logistic regression allows

discriminating each stratum in one time.

Page 14, lines 209 to 210. “Only the criterion of rash appeared to be more likely in DENV solo-infected”.

It seems also true for cervical adenopathy.

Page 18, lines 253 to 254. “Malaria diagnosis as recorded by clinicians agreed…” Agreement testing

with Cohen’s Kappa should be added in the Method section.

Table 5. For the clinical diagnosis, there is a discrepancy for Kappa measure between the text and the

table (0.636 or 0.65, respectively).

Add a new table, as Table 7 or eventually as Supplemental table for predicting absence of dengue and

absence of malaria stratum.

16 (b). Report category boundaries when continuous variables were categorized.

16 (c). If relevant, consider translating estimates of relative risk into absolute risk for a meaningful

time period

17 Other analyses

5

Report other analyses done—eg analyses of subgroups and interactions, and sensitivity analyses

Page 17, line 237. Be more explicit or the ceiling effect.

Figure 4. Why having not test inter-stratum differences and presented horizontal brackets between

tested columns and stars for setting the p value significance “|

**

|“

Discussion

The discussion is unstructured and resembles to a series of comments. Restructure the

discussion following STROBE guidelines (provide the checklist after first round of revision)

18. Key results

Summarise key results with reference to study objectives

19. Limitations

Discuss limitations of the study, taking into account sources of potential bias or imprecision. Discuss

both direction and magnitude of any potential bias

20. Interpretation

Give a cautious overall interpretation of results considering objectives, limitations, multiplicity of

analyses, results from similar studies, and other relevant evidence

21. Generalisability

Discuss the generalisability (external validity) of the study results

Conclusion

Forget Occam and his razor. Focus on take-home messages and perspectives

Discuss the paper entitled “Incidence of chikungunya virus infections among Kenyan children with

neurological disease, 2014–2018: A cohort study”, by Niamwaya DK et al (PLoS Med 2022; 19(5):

e1003994). as it is an important paper regarding the topic of the study and the study was performed

in the same country.

Other information

22. Funding

Page 25, lines 360 to 366.

Additional comments

Page 22, line 280. Write “force of infection” and not “force of transmission”. FOI is an epidemiological

parameter � = number of new infections

Number of susceptible exposed x average exposure time

All references are not conformed to PGPH standards. Revise.

6. PLOS authors have the option to publish the peer review history of their article (what does this mean?). If published, this will include your full peer review and any attached files.

**Do you want your identity to be public for this peer review?** For information about this choice, including consent withdrawal, please see our Privacy Policy.

Reviewer #1: No

Reviewer #2: **Yes: **Patrick Gérardin

---

## [Editor Report · Decision Letter 1]

30 Jun 2023

Detection of acute dengue virus infection, with and without concurrent malaria infection, in a cohort of febrile children in Kenya, 2014-2019, by clinicians or machine learning algorithms

PGPH-D-22-02073R1

Dear Dr. Vu,

We are pleased to inform you that your manuscript 'Detection of acute dengue virus infection, with and without concurrent malaria infection, in a cohort of febrile children in Kenya, 2014-2019, by clinicians or machine learning algorithms' has been provisionally accepted for publication in PLOS Global Public Health.

Best regards,

Mathieu Nacher

Academic Editor